# Deep Learning Model for Classifying and Evaluating Soybean Leaf Disease Damage

**DOI:** 10.3390/ijms25010106

**Published:** 2023-12-20

**Authors:** Sandeep Goshika, Khalid Meksem, Khaled R. Ahmed, Naoufal Lakhssassi

**Affiliations:** 1School of Computing, Southern Illinois University, Carbondale, IL 62901, USA; sandeep.goshika@siu.edu (S.G.); khaled.ahmed@siu.edu (K.R.A.); 2School of Agricultural Sciences, Southern Illinois University, Carbondale, IL 62901, USA; meksem@siu.edu

**Keywords:** deep neural networks, soybean leaf damage detection, automatic labeling, computer vision

## Abstract

Soybean (*Glycine max* (L.) Merr.) is a major source of oil and protein for human food and animal feed; however, soybean crops face diverse factors causing damage, including pathogen infections, environmental shifts, poor fertilization, and incorrect pesticide use, leading to reduced yields. Identifying the level of leaf damage aids yield projections, pesticide, and fertilizer decisions. Deep learning models (DLMs) and neural networks mastering tasks from abundant data have been used for binary healthy/unhealthy leaf classification. However, no DLM predicts and categorizes soybean leaf damage severity (five levels) for tailored pesticide use and yield forecasts. This paper introduces a novel DLM for accurate damage prediction and classification, trained on 2930 near-field soybean leaf images. The model quantifies damage severity, distinguishing healthy/unhealthy leaves and offering a comprehensive solution. Performance metrics include accuracy, precision, recall, and F1-score. This research presents a robust DLM for soybean damage assessment, supporting informed agricultural decisions based on specific damage levels and enhancing crop management and productivity.

## 1. Introduction

The imperative of sustainable food production, coupled with escalating environmental challenges and soil pollution, underscores the need to optimize farmland resources. Soybean is one of the best sources of protein (~40%), oil (~20%), and carbohydrates (~30%) in livestock diets and confers nutritional benefits contributing to diabetes and heart disease prevention [1,2,3,4]. In 2021/2022, U.S. soybean meal prices ranged from USD 300 to USD 400 per ton, from which 21 million metric tons of soymeal was used by poultry, 5.8 million metric tons by swine, 4.9 million metric tons by dairy, and 1.8 million metric tons by beef. As a vital protein source globally, soybeans are uniquely positioned to address future food security by 2050. Predominantly cultivated in major producing countries such as the USA, Brazil, Argentina, China, and India, they face challenges in meeting domestic demand for soybeans due to pathogen infections, diseases, and suboptimal agronomic practices [5,6]. While primarily used for animal feed (70%), soy finds applications in biofuels, lubricants, and other industries. Disease-related annual yield loss in the U.S. has reached nearly 11%. Diseases pose significant economic threats, exacerbated by poor cultivation practices and pathogen diversity. Recognizing diverse symptoms like anthracnose, bacterial blight, and rust is pivotal, as these diseases contribute to a 14% reduction in global food output. Production loss due to infections caused by soybean cyst nematodes (SCN) alone is estimated at more than USD 1.5 billion of dollars in the U.S. Due to the importance of this trait, several studies have investigated the SCN resistance mechanism since 1960 [7,8,9,10,11,12]. Nematodes are not the only soybean pathogens; several other significant pathogens exist across fungi, bacteria, and oomycetes, causing millions of dollars related to soybean yield loss [13,14,15,16]. Therefore, early disease detection is crucial for appropriate and fast interventions.

Traditionally, cultural practices and limited pesticide use have mitigated diseases. Manual identification by visual examination remains challenging, requiring expert intervention. The overuse of pesticides has negatively impacted the environment. Early, accurate disease detection and classification are crucial for sustainable agroecosystems. Manual assessment, while effective, needs more precision, demanding reliable technology for damage detection and evaluation. The intersection between artificial intelligence and agriculture to efficiently utilize data, improve resource management, and integrate new approaches and technologies is deemed essential to improve U.S. and global food and agriculture areas. Machine learning (ML), deep learning (DL), and computer vision (CV) have enabled rapid and accurate classification of soybean leaf disease damage severity. In this context, this research focuses on developing a system to precisely identify the extent of damage caused by soybean leaf diseases, aiding in optimal pesticide selection and crop yield enhancement. By leveraging advanced technologies, this study contributes to disease prevention, reduced pesticide use, increased product quality, and enhanced yield.

The current study aims to pioneer leaf damage assessment using computer vision and drone-mounted cameras in agriculture. While prior research has tackled healthy/unhealthy leaf detection, this study addresses the gap in quantifying damage severity. The main objectives include the following: (1) deep learning model development—create a novel deep learning model to gauge leaf damage accurately, working with images of 600 × 400 pixels or lower; (2) multiclass damage classification—categorize soybean leaf damage into five levels within a single image and optimizing computational efficiency; (3) dataset curation and training—compile a diverse dataset from Southern Illinois University farms and training the deep learning algorithm to estimate damage percentage; and (4) comprehensive evaluation—evaluate the framework’s accuracy, speed, efficiency, and offering insights into its applicability in agricultural settings. By achieving these objectives, the study aims to advance leaf damage assessment techniques, with potential implications for enhanced agricultural practices and yield management.

Traditional plant disease identification relies on visual inspection, which can be limited by observer experience and early-stage invisibility [17]. Tools like magnifying glasses can be inadequate for precise identification due to subtle variations in shape, color, and light reflections [18]. Black lights can identify some diseases, but not all, while soil fluorescence under UV light adds complexity [19]. This approach demands significant effort and expertise and may not be suitable for novices. Computer vision techniques are employed to detect soybean leaf defects, but several challenges persist. Low-quality images with noise, blur, or distortion affect algorithm performance [20]. Lighting variations introduce shadows, impacting accuracy. Background objects may lead to false positives. Image preprocessing removes noise but introduces artifacts with limited adaptability to the growth stage or leaf location.

Machine learning shows promising solutions to the above challenges but still presents some limitations. Traditional ML struggles mainly with raw data processing. Invasive techniques benefit from speed and accuracy while conventional ML requires feature expertise. Conventional ML depends on variable patterns and feature extraction, requiring repeated training. Despite successes, to the best of our knowledge, previous work has not classified damage levels [21]. Deep learning has shown promising results in various applications in agriculture, such as weed control [22,23,24,25,26], soil classification [27,28], and soil quality assessment [29,30,31]. In addition, DL demonstrates progress in identifying soybean leaf defects. DL models learn intricate patterns, aiding complex data analysis. DL eliminates manual feature engineering and scaling for large datasets and complex models [32], while transfer learning accelerates training by leveraging previous knowledge [33]. The current research employs DL for detecting soybean diseases via aerial imagery. The hybrid DL model [34] uses an optimization algorithm for soybean health in addition to detecting defect classification but not damage severity.

To our knowledge, there is a gap in deep learning-based classification of soybean leaf damage levels. This research aims to fill this void by introducing a method that collaborates deep learning and object detection, categorizing the phenotypical damage into five levels.

## 2. Results and Discussion

This section focuses on detecting and classifying damage levels in soybean leaves to enhance productivity and optimize pesticide use. The experimental setup and visualization of performance metrics for trained models on a custom dataset are outlined. Model performance, indicated by inference time, precision, and recall, is visually presented. The challenges in soybean leaf damage classification and their solutions are elaborated and discussed.

### 2.1. Environment Origin, Dataset, and Hyperparameter Tuning

The YOLOv5s model was developed using an NVIDIA Tesla V100 (32 GB) GPU, an Intel i9 11th generation CPU, 32 GB of RAM, and a 1 TB SSD to classify the level of damage in soybean leaves. The experimental setup employed the Anaconda environment with Python 3.10.9 and libraries such as Pytorch, OpenCV2, NumPy, Pillow, Matplotlib, and Pandas to train the model. The Coco data loader facilitated dataset loading and partitioning into training and validation sets. The dataset comprised 2930 images containing healthy and unhealthy soybean leaves. These images were sourced from a soybean farm at the Horticulture Research Center (Southern Illinois University) and captured using an iPhone 13 Pro camera with black and grey mats as backgrounds. Annotations followed the YOLO format, and the dataset was split into a 70% training dataset, a 20% validation dataset, and a 10% test dataset. Uniform resizing to 600 × 400 pixels was applied to all images. The different class distributions reveal that ClassOne contains nearly 3500 instances, indicating that the dataset has the highest proportion of small defects (Figure 1).

Hyperparameter tuning plays a pivotal role in optimizing model performance. Table 1 lists some hyperparameters that significantly affect optimization, such as learning rate, batch size, epochs, optimizer, input image size, regularization techniques, and anchor box dimensions. Genetic Algorithm (GA) was employed for hyperparameter optimization, given the complexity of around 30 parameters. YOLOv5’s built-in ‘evolve’ function facilitated parameter optimization. All models underwent training up to 500 epochs, utilizing the Stochastic Gradient Descent (SGD) approach for optimization [35]. The learning rate starts with lr0 and ends with lrf to improve the generalization in the YOLOv5 model (Table 1). A batch size of 16 was employed for YOLOv5 to manage memory allocation and prevent assertion errors. The Anchor_t parameter, also known as the anchor-multiple thresholds, is used to determine the maximum adjustment that can be made to the anchor boxes during training. Neurons within the neural network process input signals using the sigmoid activation function to calculate weights and facilitate information propagation. Applying the YOLOv5’s evolve function helps in finetuning and optimizing the training hyperparameters, subsequently enhancing the training outcomes.

### 2.2. Practical Analysis and Metric Comparisons

The performance evaluation of YOLOv5s ($Y_{s}$) model encompasses various metrics, including inference speed, precision, recall, mean Average Precision (mAP) at different IoU thresholds (0.5 and 0.5–0.95 with 0.05 increments), the area curves of F1, precision (P), and recall (R). The evaluation metrics for model performance are explained in depth in Section 3.7. Table 2 presents the performance metrics of the generated soybean leaf detection and classification model. Precision, recall, average precision, and mAP values are determined using Equations (3), (4), (7), and (8), respectively (Section 3.7). Inference speed is a crucial factor for damage detection efficiency, as highlighted in Table 2. The $Y_{s}$ model can classify the damage levels with an mAP of 92% while exhibiting the highest inference speed of 8.1 ms. Additionally, the results show that the $Y_{s}$ model achieved 76% mAP@0.5–0.95, 88% F1 score, 95% P curve, 97% recall, and 93% PR area under the curve. Notably, the $Y_{s}$ model achieves the highest recall and mAP at an intersection over the IoU threshold of 0.5.

The $Y_{s}$ model’s confusion matrix is demonstrated in Figure 2. Generally, a confusion matrix is utilized to show the performance of classifying the soybean leaf damage levels, in the tested $Y_{s}$ model. The center diagonal line in the confusion matrix shows the prediction results giving the soybean leave damage levels classes. In contrast, the vertical and horizontal lines show background false negative and false positive, respectively. The values on the center line range from 0 to 1, where 1 shows 100% prediction accuracy. The model performed well in predicting the soybean leaf damage classes, as shown by the values in the center diagonal line (Figure 2). These values represent the percentage of correct predictions for each class. The model achieved 89% accuracy for ClassOne, 87% for ClassTwo, 83% for ClassThree, 82% for ClassFour, and 93% for ClassFive. These results indicate that the model performed well even when the leaves had significant damage. The model was able to accurately predict the damage level for each leaf based on its features.

During the training of the $Y_{s}$ model, the classification loss was calculated as a function of epochs (Figure 3a). Figure 3a indicates that the $Y_{s}$ model was trained successfully, reaching a low loss value after 40 epochs and achieving low classification loss as it stabilizes after 450 epochs. Figure 3b shows the F1 score curve, which combines the precision and recall in one metric, varying with the confidence threshold score for $Y_{s}$ model. The performance of the $Y_{s}$ model for different soybean damage levels and confidence scores is shown in Figure 3b. The model achieved the best F1 score of 0.88 when the confidence threshold was set to 0.648, which indicates a high accuracy in classifying the soybean damage levels.

### 2.3. Testing and Validation of Trained $Y_{s}$ Model

The test dataset (~300 images) has been used to evaluate the trained model. Figure 4 shows the classification confidence results for each class of soybean leaf damage. As shown by the images, the model successfully detected and classified the damage severity for different types of leaf damage (Figure 4). The model’s performance in detecting different damage classes of soybean leaf damage is illustrated in Figure 4, where the images show how the model can identify and classify the severity of the damage.

Figure 4 shows that the generated model effectively determines whether the leaf is damaged (Figure 4a,b) or healthy (Figure 4c). Most importantly, Figure 4a,b depict that the model succeeds in detecting and classifying the damages in nonoverlapped soybean leaves, where bounding boxes and confidence scores indicate the detected instances of damage. Despite that the generated model successfully detected the level of soybean leaf damage from the five classes (mAP of 92%), detecting soybean leaf damage levels in an uncontrolled environment (field) is challenging due to leaves overlapping and different backgrounds, which still need to be elucidated.

## 3. Material and Methods

### 3.1. Object Detection and Machine Learning

The methodology revolves around identifying the optimal object detection model for damage classification. The cascaded detector, an early real-time detection approach with high accuracy, is widely used for face, pedestrian, and car detection [36]. This architecture implements sliding window detectors and has two main research streams to enhance speed: fast feature extraction and cascade learning. However, its limitation lies in constructing multiclass detectors within this design. YOLO (You Look Only Once) presents a noteworthy alternative, producing object detections within a 7 × 7 grid [22]. Despite a slight loss in detection precision, YOLO operates at approximately 40 frames per second. Combining losses on intermediate network layers enhances object identification.

Machine learning’s prowess lies in supervised learning, notably classification and regression. Learning involves improving task performance with experience “E” and performance measure “P”. Machine learning systems are categorized by factors such as human supervision, handling large data (online and batch learning), and prediction model development [37]. Supervised learning takes precedence as images are labeled to create the dataset. The proposed methodology involves labeling images to establish a labeled dataset. Consequently, this study emphasizes supervised learning techniques to determine the most effective object detection model for damage classification.

### 3.2. Dataset Creation and Annotation

Datasets are pivotal in machine learning, serving as examples for algorithms to learn from. They consist of labeled examples that guide predictions toward success or failure. Datasets are typically divided into training, validation, and test sets, training the algorithm to recognize patterns and generalize to new data. Deep learning advancements leverage data augmentation, customization, and annotation to improve model performance. This study’s dataset comprises 2930 soybean leaf images, sized at 600 × 800 pixels. It encompasses 2430 images depicting various degrees of damage and 500 healthy images sourced from Kaggle and SIU’s soybean farm. The wild-type Forrest seeds from Southern Illinois University Carbondale Agricultural Research Center were planted in the greenhouse and then transplanted in the field during summer until the end of vegetative growth as described earlier in [38]. Insect-damaged and infected leaves were collected at the V5 stage, and then pictures were taken under controlled lighting conditions using an iPhone 13 Pro camera. The images exhibit different background colors for enhanced segmentation.

Damage types include bacterial blight and defoliation. Bacterial blight lesions caused by the bacterium *Pseudomonas savastanoi* pv. *Glycinea* are angular and reddish-brown, surrounded by yellow halos [39]. As the disease progresses, lesions often grow together to produce large, irregularly shaped areas of dead tissue. The centers of older lesions frequently fall out, causing leaves to appear tattered [39]. Early symptoms of bacterial blight can be hard to distinguish from symptoms of several other diseases. It can be commonly confused with septoria brown spot, bacterial pustule, downy mildew, soybean rust, soybean vein necrosis, target spot, wildfire caused by other bacterial, oomycetes, or fungal diseases, in addition to insect and mite injury diseases. Tattered leaves can help distinguish bacterial blight, in addition to the symptoms observed in the upper canopy [39]. Defoliation refers to leaf loss due to pests, diseases, or natural causes. Notably, the dataset does not differentiate between these two symptoms, treating them as a unified condition. Figure 5 illustrates healthy leaves and diverse images with distinct damages and backgrounds of bacterial blight and defoliation.

Each image’s leaf defects are categorized into five damage levels (ClassOne to ClassFive) (Table 3), determined through a custom image processing script [40]. This dataset paves the way for model training and enhances model accuracy through specific photo labeling, a cornerstone of effective crowdsourcing strategies.

### 3.3. Automated Annotation for Image Labeling

Manual image labeling poses challenges such as time consumption, errors, and discrepancies. A custom script was developed in this research for automated labeling of leaf damage using computer vision, which drastically reduced labeling time to 50%. This script inputs leaf images and generates annotations, including bounding box coordinates, image size, total damage, image path, name, and defect name in XML format. The automated labeling process employs libraries like OpenCV, NumPy, and Matplotlib to extract image features, context, and edges. The level of damage detection relies on parameters such as leaf and damage area. For instance, to find the leaf area (Figure 6a), the edges are detected and enclosed to calculate its area (Figure 6b). Inner damages are found by identifying edges within the leaf’s outer area (Figure 6c), using morphological operations like erosion and dilation for noise reduction and edge enhancement. Automatic thresholding techniques like OTSU separate the leaf from the background. The area of a contour is calculated using the Shoelace formula (Equation (1)), considering its vertices’ coordinates. For multiple contours representing different damage areas, the percentage of damage is calculated relative to the total leaf area. The script establishes a PASCAL VOC XML format containing these annotations and calculations. Automated labeling proves efficient and consistent, mitigating human-related errors and enhancing dataset quality.
(1)$$A=0.5\;*\;\left|\left(x_{1}\;*\;y_{2}+x_{2}\;*\;y_{3}+\ldots +x_{n}\;*\;y_{1}\right)-\left(y_{1}\;*\;x_{2}+y_{2}\;*\;x_{3}+\ldots +y_{n}\;*\;x_{1}\right)\right|$$
where *n* is the number of vertices in the polygon, and $x_{1}$, $y_{1}$, $x_{2}$, $y_{2}$, …, $x_{n}$, $y_{n}$ are the coordinates of the vertices in order. The average area of the *n* leaves in the dataset is given by
$$A_{T}=\frac{A_{T_{1}}+A_{T_{2}}+A_{T_{3}}\ldots \ldots \ldots +A_{T_{n}}}{n}$$

The percentage damage of each defect in the leaf is given by
$$D_{p}=\frac{A_{T_{i}}}{A_{T}}\times c\times 10$$
where *c* is the constant used to scale up the value, $A_{T_{i}}$ is the area of defect *i*-th in the leaf.

### 3.4. Enhancing Dataset through Data Augmentation

Data augmentation techniques can mitigate challenges related to dataset size, resolution, and ground truth boxes in the collection process. Data augmentation, a vital deep learning strategy, expands the training dataset by generating new instances from existing data. This technique bolsters deep learning model performance by curbing overfitting, enhancing generalization, and augmenting data diversity [41]. YOLOv5 employs a specific augmentation technique called Mosaic augmentation (Figure 7). This method fuses four distinct images into a mosaic image, subsequently used to train the object detection model. Mosaic augmentation enhances the model’s capability to identify objects within intricate settings with multiple objects and backgrounds.

Additionally, as mosaic images replicate a range of scenarios, it diminishes the requirement for extensive training data. However, this technique can be computationally intensive, demanding the creation of numerous images and labels per training batch. Proper hyperparameter adjustments, like crop patch size and overlap, are essential to ensure effective learning from mosaic images. To bolster deep learning model efficacy in scenarios with limited or imbalanced data, several strategies like geometric transformations, color space transformations, kernel filters, and meta-learning can be implemented for image augmentation [41]. Data augmentation generates new data akin to the original dataset but with slight variations [42]. Techniques include cropping, flipping, rotating, blurring, scaling, translating, color perturbations, noise addition, and more. By embracing these augmentation strategies, the model learns from an enriched dataset, surmounting data scarcity and imbalance constraints.

### 3.5. Model Architecture and Training

Challenges related to dataset size, resolution, and ground truth boxes in the collection process can be mitigated using data augmentation techniques. Data augmentation, a vital deep learning strategy, expands the training dataset by generating new instances from existing data. This technique bolsters deep learning model performance by curbing overfitting, enhancing generalization, and augmenting data diversity [41].

The YOLOv5, a convolutional neural network (CNN)-based object detector, offers advantages like high accuracy, rapid detection speed, and lightweight nature [43]. Object detectors of this kind are classified into different categories based on their attributes and features. These detectors primarily consist of two components: a CNN-based backbone for image feature extraction and a detection head to predict object classes and bounding boxes. The YOLOv5 object detector incorporates intermediate layers, the neck of the detector, positioned between the backbone and the head, as depicted in Figure 8.

Backbone: YOLOv5 employs a CSPDarknet53 backbone, a modified version of Darknet53, known for its efficiency in various computer vision tasks. Cross-stage partial connections (CSP) are integrated into CSPDarknet53 to streamline information flow and reduce parameter count, enhancing its efficiency compared to the original Darknet53 [44]. The essential component of this backbone is the CBS module, which consists of a Convolution, BatchNorm, and SiLu activation function [45,46]. They stacked together and formed the C3 module. Additionally, the YOLOv5 model includes supplementary layers like Spatial Pyramid Pooling Fast (SPP) (https://github.com/ultralytics/yolov5, accessed on 12 June 2023) to boost its performance further [47]. The SSPF module includes several maxpooling layers and a CBS module. These modules fuse the feature maps of different receptive fields and enrich the expression ability of feature maps. Each Concat layer is used to slice the previous layer. This backbone demonstrates robust feature extraction capabilities for classification and other tasks, often fine tuned for versatility across activities.

Neck: The neck, a pivotal component in the object detection framework, enhances the utilization of features extracted by the backbone. It optimally processes and employs the feature maps extracted by the backbone across various stages. The YOLOv5 neck also incorporates up-sampling and down-sampling stages, combining CSPDarknet53 feature maps using C3 to create the pyramid. Additionally, lateral connections enable feature propagation across scales. The neck outputs feature maps of 80 × 80 × 256, 40 × 40 × 512, and 20 × 20 × 1024 that correspond to target objects of a small, medium, and large scale.

Head: The head encompasses the final 3 Conv2D layers of the network, which are responsible for generating model output. While the backbone focuses on classification, the head manages object positioning tasks, determining object likelihood within bounding boxes and their respective categories. The head employs feature maps extracted from the backbone to estimate bounding box offsets relative to predefined anchor boxes, enhancing model accuracy [47].

### 3.6. YOLOv5 Architecture and Analysis

YOLOv5 is a one-stage object detector, structured with 1 focus layer, 10 backbone layers, and 14 neck layers. For instance, “Conv (256)” is the third convolutional layer, taking 128 pixels as input and producing 256 pixels as output for the next layer. The backbone extracts feature information, directing it to specialized convolutional layers responsible for object prediction and detection. SPPF generates fixed-size windows, independent of input image dimensions, enhancing damage detection in soybean leaves. Max pooling with kernel lengths 5, 9, and 13 contributes to this process. The last convolutional layer in the backbone bridges the head and backbone, creating feature pyramids for object scaling. The model architecture includes customizable particular convolutional layers. For the level of damage detection in soybean leaves, layers 17, 20, and 23 are selected with 256, 512, and 1024 pixels, respectively, and used to determine object classes with kernel stride.

The YOLOv5 model builds upon its predecessors, combining elements from YOLO to YOLOv4. It divides the image into S∗S grids, each responsible for detecting the presence of objects (soybeans), including leaf damage. *β* bounding boxes predict the extent of leaf damage, each associated with a confidence score. Each grid cell supports multiple bounding box predictions characterized by x, y, w, h, and c attributes. The model’s output computation for a single image involves S∗S∗β∗5 calculations. The model determines the level of damage in the soybean leaf depending on confidence scores. Speed and accuracy are prioritized, influencing activation optimizations and regularization loss values. Model loss is a summation of bounding box loss and classification loss.

Activation layers employ the SiLU (Sigmoid Linear Unit) activation function, while SGD (Stochastic Gradient Descent) serves as the optimization function. Pytorch is the model design and detection platform, with binary cross-entropy as the loss function. Anchor boxes are applied to features, generating output vectors containing class probabilities, object scores, and bounding boxes.

### 3.7. Evaluation Metrics for Model Performance

Evaluation metrics are crucial in assessing a deep learning model’s performance on a specific task. These metrics provide insights into the quality of the model’s predictions and guide improvements. Statistical methods, such as confusion matrix, F1 curve, precision, recall, accuracy, and GIoU (Generalized Intersection over Union), are commonly employed for model evaluation.

Parameters used in detection and evaluation include T_P_: True Positive, T_N_: True Negative, F_N_: False Negative, and F_P_: False Positive.

A correct prediction of the object’s level of damage to the soybean leaf within the bounding box is categorized as a true positive. Specifically, if the Intersection over Union (IoU) between the predicted and actual bounding boxes is greater than or equal to a defined threshold (denoted by α), it is considered a true positive. Conversely, a frame is labeled as a false positive when the object is absent from the bounding box and IoU is less than α. If the bounding box fails to capture the target object, it results in a false negative. True negative is assigned when the object is correctly absent or not predicted in the frame.

Model accuracy hinges on the correctness of its detections in unseen data. The following evaluation metrics are utilized in this research to assess the model’s performance:

IoU: Intersection of Union is a term used to define the extent of overlap of two bounding boxes. The greater the region of overlap, the greater the IOU. It is given by the ratio area of the intersection of two boxes to the area of the union of two boxes.
(2)$$IoU=\frac{A\cap B}{AUB}$$
where A, B predicted and ground truth boxes.

GIoU: Generalized IoU (GIoU) is an extension of the Intersection over Union (IoU). This metric explains the size and location of the predicted bounding box and the ground truth bounding box, where C is the smallest convex shape that encloses both boxes A and B. We calculate a ratio between the area occupied by C excluding A and B and divide by the total area occupied by C. Finally, GIoU is calculated by subtracting this ratio from the IoU value as shown in Equation (3) [48].
(3)$$GIoU=IoU-\frac{\left|C\backslash \left(A\cup B\right)\right|}{\left|C\right|}$$

Recall score is calculated as the ratio of true positives (T_P_) to the sum of true positives and false negatives (F_N_).
(4)$$R=\frac{T_{P}}{T_{P}+F_{N}}$$

Precision is the ratio of true positives to the sum of true positives and false positives.
(5)$$P=\frac{T_{P}}{T_{P}+F_{P}}$$

Accuracy is the ratio of the number of correct predictions (true positives and true negatives) to the total number of predictions T_P_, T_N_, F_P_, and F_N_.
(6)$$A=\frac{T_{P}+T_{N}}{T_{P}+T_{N}+F_{P}+F_{N}}$$

$F_{1}$ Curve is the harmonic mean of precision and recall, and it is a single number that summarizes the performance of the model across all possible classification thresholds.
(7)$$F_{1}=\frac{2*P*R}{P+R}$$

The area under the precision–recall curve is calculated using numerical integration, which gives the AP score.
(8)$$AP@\alpha =\int_{0}^{1}p\left(r\right)dr$$

Mean Average Precision is the average of the AP@*α* calculated for all the classes over IoU Threshold *α* depending on the detection challenge.
(9)$$mAP@\alpha =\frac{1}{N}\sum_{i=1}^{N}AP_{i}$$

The loss function is used to measure the difference between the predicted output of a model and the true output, and it is given by the summation of coordination loss, classification loss, and object loss, as follows.
(10)$$\lambda_{coord}\sum_{i=0}^{S^{2}}\sum_{j=0}^{\beta }l_{ij}^{obj}{\left(x_{i}-{\hat{x}}_{i}\right)}^{2}+{\left(y_{i}-{\hat{y}}_{i}\right)}^{2}$$

Equation (9) is used to find the loss of the bounding box of position (*x*, *y*) and the actual position ($\hat{x},\hat{y}$) from the training data. This function computes a sum over each bounding box predictor (*j* = 0…*β*) of each grid cell (*i* = 0…$S^{2}$)$l_{ij}^{obj}$ where $l_{ij}^{obj}$ implies that the object appears in cell *i* and indicates that jth bounding box predictor in cell *i* is responsible for that prediction [48].

The coordination error calculated by the prediction box w.r.t width/height is given below in Equation (11).
(11)$$\lambda_{coord}\sum_{i=0}^{S^{2}}\sum_{j=0}^{\beta }l_{ij}^{obj}\left[{\left(\sqrt{w_{i}}-\sqrt{\hat{W_{i}}}\right)}^{2}+{\left(\sqrt{h_{i}}-\sqrt{\hat{h_{i}}}\right)}^{2}\right]$$

The classification loss based on the confidence score for each bounding box class is given by Equation (11).
(12)$$\lambda_{coord}\sum_{i=0}^{S^{2}}\sum_{j=0}^{\beta }l_{ij}^{obj}{\left(C_{i}-{\hat{C}}_{i}\right)}^{2}+\lambda_{noobj}\sum_{i=0}^{S^{2}}\sum_{j=0}^{\beta }l_{ij}^{noobj}{\left(C_{i}-{\hat{C}}_{i}\right)}^{2}\;$$

The Equation (11) is used to initialize the *λ* value for the presence of an object differently to gain model stability or the presence of an object $\lambda_{coord}$ = 5, else $\lambda_{noord}$ = 0.5.

## 4. Conclusions and Future Work

The necessity of accurately assessing the degree of damage in soybean leaves for enhanced production and reduced disease susceptibility has prompted the utilization of neural networks. Training a neural network from the ground up aimed to ascertain the extent of leaf damage and subsequently compare outcomes against a compiled dataset. Various augmentation techniques were implemented to amplify the data size, encompassing image blurring, rotation, and resolution adjustments. Experimental findings highlight the $Y_{s}$ model’s commendable performance in terms of mAP@0.5. In extending this research, several avenues for improvement emerge. Firstly, there is a demand to expand the dataset by capturing a broader range of samples with overlapped leaves, encompassing diverse forms of soybean leaf damage. Furthermore, transitioning from solely detecting leaf damage to developing a neural network capable of automatically identifying overall plant defects emerges as a promising avenue for further exploration.

## Figures and Tables

**Figure 1 ijms-25-00106-f001:**
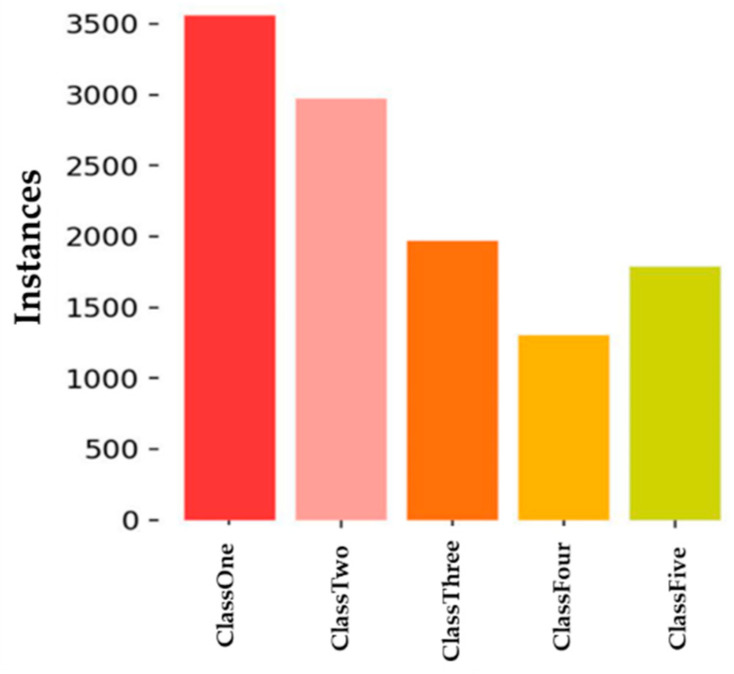
Graphical representation of instances of each class corresponding to the collected soybean leaf damage.

**Figure 2 ijms-25-00106-f002:**
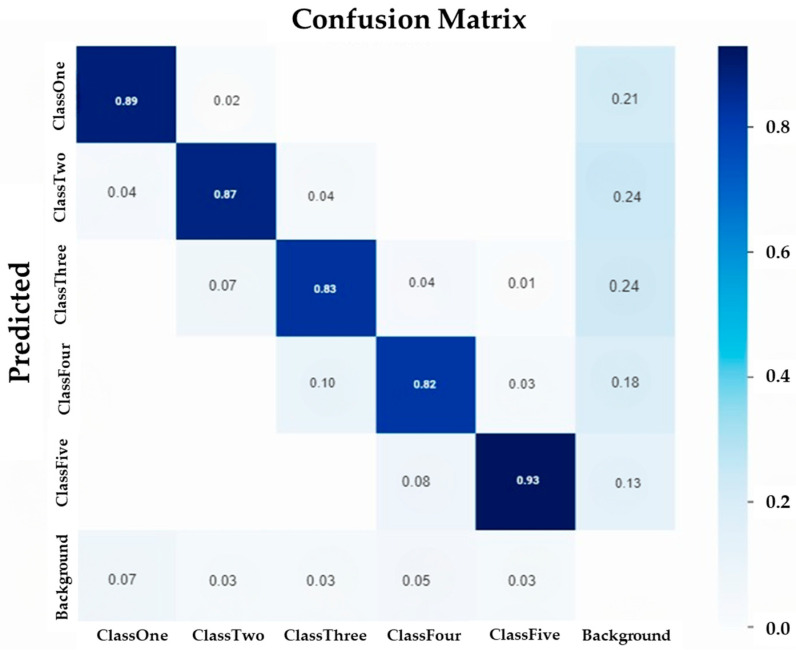
Confusion matrix of the YOLOv5 model in the test set (10% of the data). Percentages of correctly classified soybean damage classes fall on the diagonal. However, percentages that fall off the diagonal correspond to incorrectly classified classes.

**Figure 3 ijms-25-00106-f003:**
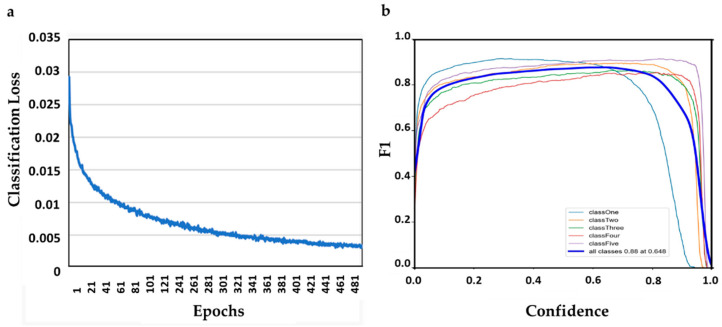
(**a**) represents the classification loss of the YOLOv5 model and (**b**) illustrates the YOLOv5s model’s F1 score curves corresponding to the five damage levels classes.

**Figure 4 ijms-25-00106-f004:**
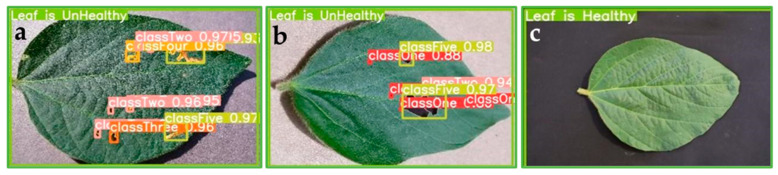
Model detection of damage to leaves. (**a**) Model detection of unhealthy leaf, (**b**) model detection of unhealthy leaf, and (**c**) model detection of healthy leaf.

**Figure 5 ijms-25-00106-f005:**
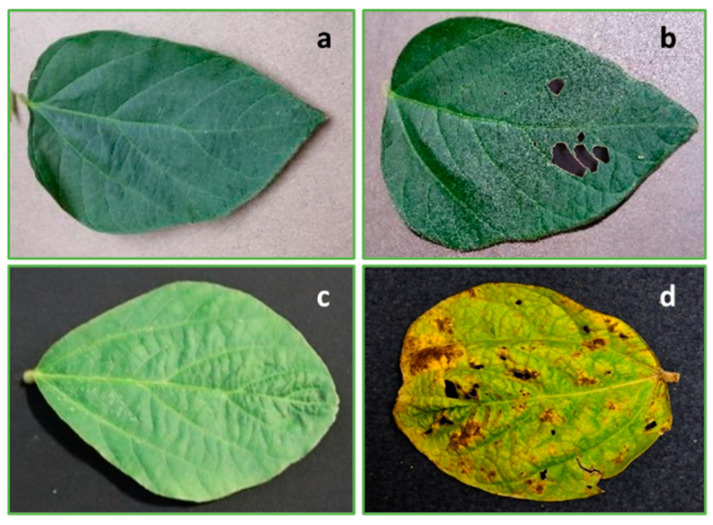
Sample images from the dataset: (**a**) healthy leaf with brown color as background, (**b**) defoliation of leaf with brown color as background, (**c**) healthy leaf with black color as background, and (**d**) yellow disease leaf with black color as background.

**Figure 6 ijms-25-00106-f006:**
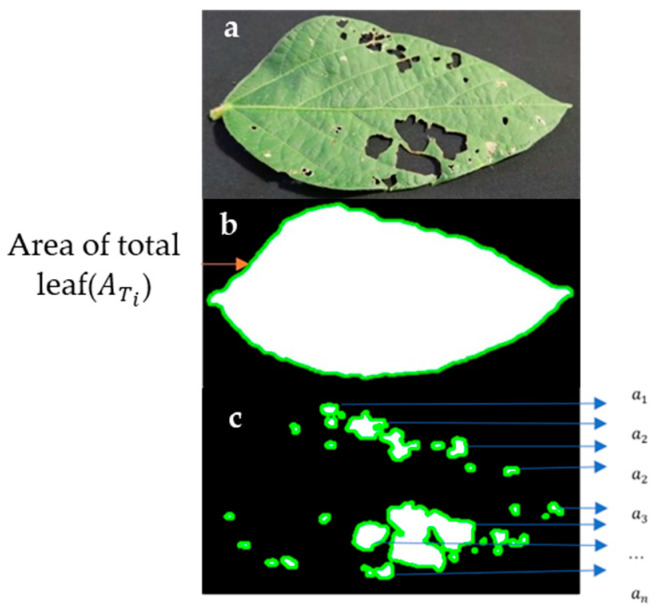
Original leaf images and converted binary images were used to draw contours. (**a**) Original image with damage, (**b**) total leaf area with contours, and (**c**) inner damage of leaf with contours.

**Figure 7 ijms-25-00106-f007:**
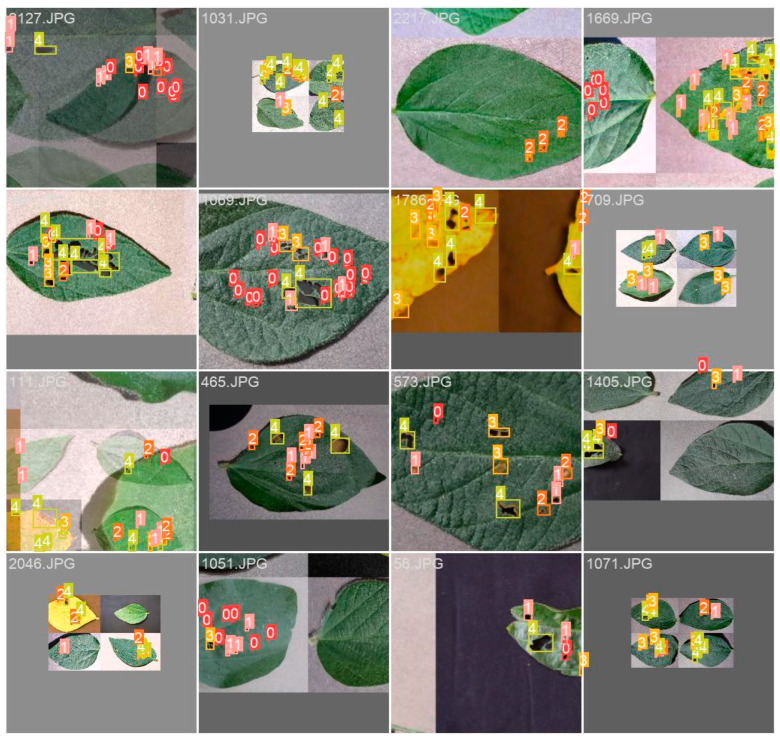
An example of Mosaic augmentation generated by YOLOv5 for the soybean dataset. It includes more diverse soybean leaves and backgrounds.

**Figure 8 ijms-25-00106-f008:**
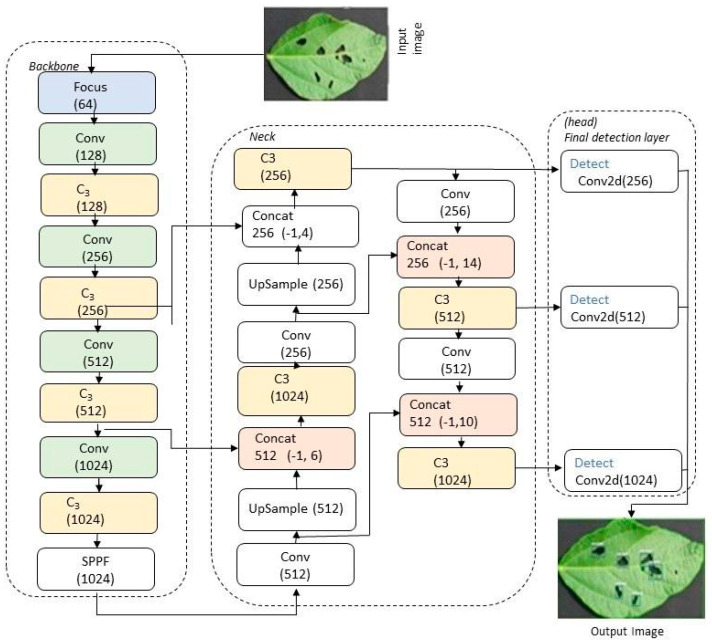
Architecture of the YOLOv5 neural network that includes 10 backbone layers, 14 neck layers, and 3 Conv2d layers in the head.

**Table 1 ijms-25-00106-t001:** Values of the hyperparameters.

Hyperparameter	Value
Learning rate	lr0: 0.00334–lrf: 0.15135
Weight decay	0.00025
Batch size	16
Epochs	500
Optimizer	SGD
Input image size	600 × 800
Anchor_t	4

**Table 2 ijms-25-00106-t002:** Performance metrics of soybean leaf damage detection and classification model.

Performance Metric	$\mathrm{Model}\;(Y_{s}$)
Inference Speed (ms)	8.1
Precision (%)	88
mAP@0.5 (%)	92
mAP@0.5–0.95 (%)	76
$F_{1}$ Curve (%)	88
P Curve (%)	95
R Curve (%)	97
PR Curve (%)	93

**Table 3 ijms-25-00106-t003:** Classification and the percentage damage in soybean leaf.

Classification	Percentage Damage of Each Defect in Leaf $\;(D_{p})$
ClassOne	Greater than 0 and less than 1.1
ClassTwo	Greater than 1.1 and less than 2.4
ClassThree	Greater than 2.4 and less than 4.1
ClassFour	Greater than 4.1 and less than 6.6
ClassFive	Greater than 6.7

## Data Availability

The data presented in this study are available on request from the corresponding author.

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
