# Peer review of "Deep Learning Model for Classifying and Evaluating Soybean Leaf Disease Damage"

_ijms, 2023, doi:10.3390/ijms25010106_

Round 1

Reviewer 1 Report

Comments and Suggestions for Authors

The paper discusses the challenges faced by soybean crops, which are influenced by various factors such as pathogen infections, environmental changes, poor fertilization, and incorrect pesticide application, resulting in reduced yields. To address this issue, the study introduces a novel Deep Learning Model (DLM) that has been trained on a substantial dataset of nearly 2,930 soybean leaf images. This DLM goes beyond traditional binary healthy/unhealthy leaf classification and is capable of predicting and categorizing soybean leaf damage severity into five levels. It offers a comprehensive solution for tailored pesticide application and yield projections. The model's performance is evaluated using metrics like accuracy, precision, recall, and F1-score. The research presents a robust DLM for assessing soybean damage, enabling more informed agricultural decisions based on specific damage levels, ultimately enhancing crop management and productivity. The method developed is highly interesting, well-substantiated, and clearly applicable. Nevertheless, there appears to be a deficiency in the discussion of the presented results. The interpretation of the data in Table 2 and Figure 6 is not sufficiently elucidated. Furthermore, the captions are overly simplistic and do not effectively convey the results to the reader. I recommend enhancing the results section to provide a more comprehensive explanation and discussion of the data. Additionally, there are some minor points that need to be addressed:

Line 8: Glycine max is in italic.

Lines 39-40: “Anthracnose”, “Bacterial Blight”, and “Rust” must be in lower case.

Line 43: SCN is “soybean cyst nematodes”? All abbreviation must be indicated in the text.

Line 56: “Learning”, “Deep Learning”, and “Computer Vision” must be in lower case.

Lines 68-69: “Curation”, “Training”, and “Compile” must be in lower case.

Lines 70-71: “Evaluation” and “Evaluate” must be in lower case.

Line 72: “By” must be in lower case.

Line 86: The abbreviation form “ML” must be indicated the first time “machine learning” is cited in the manuscript. “Machine learning” was cited in the first time in line 56.

Line 90: The abbreviation form “DL” must be indicated for the first time in line 56.

Lines 139, 141, and 147: Remove the yellow highlight of reference 39.

Lines 220 and 225: What is the correct form, “Yolov5” or “YOLOv5”? This needs to be standardized.

Line 225: What is CNN? It need to be indicated in the text.

Line 392: add space at the beginning of the sentence

Figure 2: To enhance the visual appeal of the image, it is crucial to ensure precise alignment of the subfigures.

Figure 3: The figure is very small and has poor definition. It is impossible for the reader to see it.

Figure 4: The figure need to a higher definition.

Figure 6: The characters in the figure need to be larger. The legend is incomplete and needs to provide more information about the figure.

Author Response

Reviewer 1: Comments and Suggestions for Authors

The paper discusses the challenges faced by soybean crops, which are influenced by various factors such as pathogen infections, environmental changes, poor fertilization, and incorrect pesticide application, resulting in reduced yields. To address this issue, the study introduces a novel Deep Learning Model (DLM) that has been trained on a substantial dataset of nearly 2,930 soybean leaf images. This DLM goes beyond traditional binary healthy/unhealthy leaf classification and is capable of predicting and categorizing soybean leaf damage severity into five levels. It offers a comprehensive solution for tailored pesticide application and yield projections. The model's performance is evaluated using metrics like accuracy, precision, recall, and F1-score. The research presents a robust DLM for assessing soybean damage, enabling more informed agricultural decisions based on specific damage levels, ultimately enhancing crop management and productivity. The method developed is highly interesting, well-substantiated, and clearly applicable.

Nevertheless, there appears to be a deficiency in the discussion of the presented results.

Response: We would like to thank the reviewer for his recommendations and suggestions to improve the manuscript. We have made the requested edits and modifications as suggested by the reviewer.

The interpretation of the data in Table 2 and Figure 6 is not sufficiently elucidated. Furthermore, the captions are overly simplistic and do not effectively convey the results to the reader. I recommend enhancing the results section to provide a more comprehensive explanation and discussion of the data.

Response: This section has been updated as suggested by the reviewer. Please see the new edits on the attached version of the manuscript on pages 3, 4, 5, and 6.

Additionally, there are some minor points that need to be addressed:

Line 8: Glycine max is in italic.

Response: This has been corrected.

Lines 39-40: “Anthracnose”, “Bacterial Blight”, and “Rust” must be in lower case.

Response: This has been corrected. 

Line 43: SCN is “soybean cyst nematodes”? All abbreviation must be indicated in the text.

Response: This has been corrected. Soybean cyst nematode was first cited and abbreviated in line 42.  

Line 56: “Learning”, “Deep Learning”, and “Computer Vision” must be in lower case.

Response: This has been corrected.

Lines 68-69: “Curation”, “Training”, and “Compile” must be in lower case. This has been corrected.  Response: Done

Lines 70-71: “Evaluation” and “Evaluate” must be in lower case.

Response: This has been corrected.

Line 72: “By” must be in lower case. This has been corrected.

Response: Done

Line 86: The abbreviation form “ML” must be indicated the first time “machine learning” is cited in the manuscript. “Machine learning” was cited in the first time in line 56. This has been corrected.

Response: Done

Line 90: The abbreviation form “DL” must be indicated for the first time in line 56.

Response: This has been corrected.

Lines 139, 141, and 147: Remove the yellow highlight of reference 39.

Response: The highlight has been removed.

Lines 220 and 225: What is the correct form, “Yolov5” or “YOLOv5”? This needs to be standardized.

Response: This has been corrected to: “YOLOv5”.

Line 225: What is CNN? It needs to be indicated in the text.

Response: CNN has been indicated in the text as suggested by the reviewer.

Line 392: add space at the beginning of the sentence.

Response: This has been corrected.

Figure 2: To enhance the visual appeal of the image, it is crucial to ensure precise alignment of the subfigures.

Response: The alignment of the figure has been adjusted.

Figure 3: The figure is very small and has poor definition. It is impossible for the reader to see it.

Response: We improved the resolution and size of Figure 3.

Figure 4: The figure needs to a higher definition.

Response: We improved the resolution of Figure 4 and all the rest of the figures.

Figure 6: The characters in the figure need to be larger. The legend is incomplete and needs to provide more information about the figure.

Response: We enlarged the characters, improved the resolution of Figure 6, and provided a complete figure legend.

Reviewer 2 Report

Comments and Suggestions for Authors

This paper develops a deep learning approach to automatically detect and quantify the level of disease in isolated soybean leaves. It is based on YOLOv5 and produces bounding boxes of the infected areas of the leaves (cf. Figure 3) along with a quantification (one of 5 levels of degree of leaf damage). The method is explained and then tested to obtain quantitative measures of performance. As summarized in Fig. 6, the success rate of classification in those 5 classes is above 80%, a good result.

The training was done on images of collected leaves put flat on a uniform background, making segmentation quite easy and lowering image variation. The validation was done on similar images (with uniform backgrounds) so it is unclear how useful the authors' approach is in natural settings where the leaf is still on the plant and the background is uncontrolled. Also, they have 30 hyperparameters but give no details about how important those are. (See for example my request in Detailed points to see how performance is affected by threshold values.) I recommend the authors have Supplementary Material to show to what extent the performances change under different runs of the hyperparameter optimizer unless it converges quite generally to the same local optimum.  

Separate comments:
1: The English is somewhat clumsy but that is not an impediment to reading and understanding the text. (But do see what you can do to improve the sentences, e.g. using ChatGPT.)
2: You use 5 classes. Wouldn't it have been more natural to just output the proportion of the leaf that is damaged?

Detailed points
- line 29: is "$400/metric ton" referring to the price? (If so, rephrase as the reader is otherwise led to think you are referring to the mass produced but then the numbers do not make sense.) Suggestion: "In 2021/2022, of the production of US soymeal, with an average cost of $400 per metric ton, 21 million metric tons were used ..."
- line 42: billions of dollars?
- line 43: put SCN in full text the first time.
- line 131: until the end of vegetative growth?
- lines 138 to 141: duplicated sentence.
- line 186: why "first"? In the formula, you want the sum of all a_i, no?
- Figure 3: you might want to separate better the different sub-images to make cleare you have 4 leaves per sub-image when that is the case.
- Figure 4: what is the input of XVIII Conv?
- Definition of GIoU: you don't say what C is (and mention the definition of "/"). Also put in bold the other defined terms (beyond IoU and GIoU).
- Figure 5: there are only 2930 images so mention somewhere that there are multiple leaves per image. Do you have overlaps between leaves in the pictures or are the leaves always non overlapping, facilitating segmentation into the correct number of leaves?
- In table 2 (the benchmarks for the threshold IoU = 0.5) you should include data for other threshold values so we know how sensitive these performances are. Or better, give results as curves for the different metrics as a function of that threshold.
- Check the first vs last name of some references and that you have not left out some of the associated authors, while there are other references that are duplicated.

Comments on the Quality of English Language

Should be improved (but not a major issue).

Author Response

Reviewer 2:

Comments and Suggestions for Authors

This paper develops a deep learning approach to automatically detect and quantify the level of disease in isolated soybean leaves. It is based on YOLOv5 and produces bounding boxes of the infected areas of the leaves (cf. Figure 3) along with a quantification (one of 5 levels of degree of leaf damage). The method is explained and then tested to obtain quantitative measures of performance. As summarized in Fig. 6, the success rate of classification in those 5 classes is above 80%, a good result.

Response: We would like to thank the reviewer for his recommendations and suggestions to improve the manuscript. We have made the requested edits and modifications as suggested by the reviewer.

The training was done on images of collected leaves put flat on a uniform background, making segmentation quite easy and lowering image variation. The validation was done on similar images (with uniform backgrounds) so it is unclear how useful the authors' approach is in natural settings where the leaf is still on the plant and the background is uncontrolled.

Response: Thank you to the reviewer for mentioning this. We are planning to investigate this aspect on our future work in order to develop a model that can quantify the damage degree in natural settings where leaves may overlap. We are planning to plant several thousand of soybeans at the Horticulture Research Center (SIUC) during the summer 2024, use a drone to take pictures, and then use the developed approach to detect leaf damage. The scope of the work was introducing a novel DLM for accurate damage prediction and classification by training a high number of near-field soybean leaf images. The developed model successfully quantifies damage severity, distinguishing healthy/unhealthy leaves and offering a comprehensive solution in precision agriculture. The outcome of this research presents a robust DLM for soybean damage assessment, supporting informed agricultural decisions based on specific damage levels and enhancing crop management and productivity.

Also, they have 30 hyperparameters but give no details about how important those are. (See for example my request in Detailed points to see how performance is affected by threshold values.)

Response: We would like to thank the reviewer for catching this. Setting hyper-parameters in training deep learning models is challenging. Therefore, as we mentioned in the paper (lines 363-365), we used a genetic algorithm that has a built-in evolve function developed in YOLOV5. The evolve function optimized the hyperparameters used in the current study.

I recommend the authors have Supplementary Material to show to what extent the performances change under different runs of the hyperparameter optimizer unless it converges quite generally to the same local optimum.

Response: We used the YOLOv5's built-in 'evolve' function, which is based on a genetic algorithm for parameter optimization. This function converges to the best hyperparameter that we used in training the model. We have summarized the hyperparameters and their obtained values in Table 1.

Separate comments:
1: The English is somewhat clumsy but that is not an impediment to reading and understanding the text. (But do see what you can do to improve the sentences, e.g. using ChatGPT.)

Response: We have improved the reading of the manuscript.

2: You use 5 classes. Wouldn't it have been more natural to just output the proportion of the leaf that is damaged?

Response: We would like to thank the reviewer for this reflection. Unlike traditional programming, which relies on predefined rules and solutions to produce specific outputs. Machine learning (ML) and deep learning (DL) are data-driven approaches that learn from examples/data and generate models that can handle unseen data and scenarios. For instance, instead of writing a program to calculate the exact percentage of leaf damage based on fixed criteria, we trained an ML/DL model to learn from a large dataset of leaf images and labels. Then, we used the model to predict the confidence score of new leaf images belonging to one of the five classes we defined in Table 3.

Detailed points
- line 29: is "$400/metric ton" referring to the price? (If so, rephrase as the reader is otherwise led to think you are referring to the mass produced but then the numbers do not make sense.) Suggestion: "In 2021/2022, of the production of US soymeal, with an average cost of $400 per metric ton, 21 million metric tons were used ..." Response: This has been updated (lines 29-30).
- line 42: billions of dollars?  Response: This has been corrected.
- line 43: put SCN in full text the first time. Response: This has been updated.            
- line 131: until the end of vegetative growth? Response: This has been updated.      
- lines 138 to 141: duplicated sentence. Response: The duplicated sentence has been deleted.                
- line 186: why "first"? In the formula, you want the sum of all a_i, no? Response: Thank you for catching this typo. It should be “a_i”. We corrected as follow:  is the area of defect i-th in the leaf.

- Figure 3: you might want to separate better the different sub-images to make clear you have 4 leaves per sub-image when that is the case.

Response: Figure 3 (updated figure 7) shows a Mosaic image which is automatically generated by YOLOv5, where four or more random images are combined into four tiles with a random ratio with a relative scale of the image and trained through a data loader.

- Figure 4: what is the input of XVIII Conv?

Response: The model Figure 4 (updated see Figure 8) removed the roman numbers that indicate levels in the YOLOv5 model to avoid ambiguity. Conv is an abbreviation for a convolutional neural network, which is used to process leaf images for feature extraction.

- Definition of GIoU: you don't say what C is (and mention the definition of "/"). Also put in bold the other defined terms (beyond IoU and GIoU).

Response: We updated the paper by defining “C” and we have explained the equation 2 and added the appropriate reference.

- Figure 5: there are only 2930 images so mention somewhere that there are multiple leaves per image. Do you have overlaps between leaves in the pictures or are the leaves always non overlapping, facilitating segmentation into the correct number of leaves?

Response: We have clarified that we have used a non-overlapping set of images.

- In table 2 (the benchmarks for the threshold IoU = 0.5) you should include data for other threshold values so we know how sensitive these performances are. Or better, give results as curves for the different metrics as a function of that threshold.

Response: Table 2 includes a performance metric [email protected], which shows the mean precision for threshold values ranging from 0.5 to 0.95 with 0.05 increments.

- Check the first vs last name of some references and that you have not left out some of the associated authors, while there are other references that are duplicated.

Response: We have used endnote to insert all references.
